# Hydroxychloroquine Effects on THP-1 Macrophage Cholesterol Handling: Cell Culture Studies Corresponding to the TARGET Cardiovascular Trial

**DOI:** 10.3390/medicina58091287

**Published:** 2022-09-16

**Authors:** Saba Ahmed, Justin Konig, Lora J. Kasselman, Heather A. Renna, Joshua De Leon, Steven E. Carsons, Allison B. Reiss

**Affiliations:** Department of Medicine and Biomedical Research Institute, NYU Long Island School of Medicine, Mineola, NY 11501, USA

**Keywords:** atherosclerosis, cholesterol, hydoxychloroquine, macrophage, scavenger receptors

## Abstract

*Background and Objectives*: Cardiovascular (CV) risk is elevated in rheumatoid arthritis (RA). RA patient plasma causes pro-atherogenic derangements in cholesterol transport leading to macrophage foam cell formation (FCF). The TARGET randomized clinical trial compares CV benefits of 2 RA drug regimens. Hydoxychloroquine (HCQ) is a key medication used in TARGET. This study examines effects of HCQ on lipid transport to elucidate mechanisms underlying TARGET outcomes and as an indicator of likely HCQ effects on atherosclerosis in RA. *Materials and Methods*: THP1 human macrophages were exposed to media alone, IFNγ (atherogenic cytokine), HCQ, or HCQ + IFNγ. Cholesterol efflux protein and scavenger receptor mRNA levels were quantified by qRT-PCR and corresponding protein levels were assessed by Western blot. FCF was evaluated via Oil-Red-O and fluorescent-oxidized LDL. Intracellular cholesterol and efflux were quantified with Amplex Red assay. *Results*: With the exception of a decrease in the efflux protein cholesterol 27-hydroxylase in the presence IFNγ at all HCQ concentrations, no significant effect on gene or protein expression was observed upon macrophage exposure to HCQ and this was reflected in the lack of change in FCF and oxidized LDL uptake. *Conclusions*: HCQ did not significantly affect THP1 macrophage cholesterol transport. This is consistent with TARGET, which postulates superior effects of anti-TNF agents over sulfasalazine + HCQ.

## 1. Introduction

Rheumatoid Arthritis (RA) is one of the most common autoimmune disorders and a worldwide public health challenge affecting approximately 1% of the population globally [1,2]. Women between the ages of 40–60 are disproportionately affected [3,4]. Primary manifestations of RA involve joint pain, swelling and tenderness. Gradually, uncontrolled inflammation leads to disease progression, irreversible destruction of cartilage and bone, reduced range of motion, and severe physical disability [5]. One of the most compelling clinical challenges in the management of RA is the development of atherosclerotic cardiovascular disease (ASCVD) [6,7]. It is well-established that RA confers increased risk for dyslipidemia and atherosclerosis [8,9,10]. RA patients have an approximately 50% higher risk of ASCVD than the general population and 1.6 times higher rate of acute myocardial infarction and ischemic stroke [6,7]. The substantial burden ASCVD places on RA patients, contributes to overall morbidity, mortality, and decline in physical function with diminished quality-of-life [11,12]. Despite these clear risks to heart health, guidelines for the management of ASCVD in the RA patient are scarce. The atheroprotective or atherogenic properties of RA treatments have not been adequately characterized. The TARGET (Treatments Against RA and Effect on 18-Fluorodeoxyglucose [18 F-FDG] Positron Emission Tomography [PET]/Computed Tomography [CT]) trial (NCT02374021) seeks to remedy this situation by comparing extent of vascular inflammation in 150 RA subjects with inadequate response to methotrexate randomized to add either a tumor necrosis factor (TNF) inhibitor (etanercept or adalimumab) or sulfasalazine and hydroxychloroquine (HCQ) to the methotrexate [13].

Our in vitro study focuses on the HCQ component of TARGET and its potential as an atheroprotectant. HCQ, commonly prescribed for the treatment of RA, exerts therapeutic effects as an immunomodulator and immunosuppressive with anti-inflammatory and antioxidant activity [14,15,16]. The mechanisms of action of HCQ are not completely understood; however, anti-inflammatory activity is attributed to downregulation of cytokine production and secretion by monocytes and T cells [17]. Reduced cytokine production may occur due to HCQ inhibition of toll-like receptor pathways. HCQ may also exert anti-inflammatory effects by increasing the pH of endosomal compartments of lysosomes. This would interfere with the normal participation of the lysosomal pathway in antigen processing and Major Histocompatibility Complex (MHC) class II-mediated antigen presentation [18]. HCQ may also improve the lipid profile in RA, reducing low-density lipoprotein (LDL) cholesterol [19].

As we await the results of TARGET, our study looks at underlying mechanisms of HCQ action on macrophages, the key cell type involved in lipid handling and foam cell formation in atherosclerosis. Ultimately, our complementary approaches seek to determine whether HCQ is clinically effective in ASCVD prevention in RA and the reasons underlying the outcome.

## 2. Materials and Methods

### 2.1. Cell Culture and Experimental Conditions

THP-1 monocytes (American Type Culture Collection, Manassas, VA, USA) were cultured in a complete growth medium (RPMI-C) of RPMI 1640 supplemented with 10% fetal bovine serum (FBS; Gibco), 1% penicillin-streptomycin, and 1% glutamate at 37 °C in a 5% CO2 atmosphere to a density of 106 cells per ml.

After initial plating, monocytic THP-1 cells were differentiated into adherent macrophages with 100 nM phorbol 12-myristate 13-acetate (PMA), obtained from Sigma-Aldrich (St. Louis, MO, USA), for 18–24 h at 37 °C. Once the differentiated phenotype had been achieved, the PMA-containing medium was removed and replaced with complete RPMI 1640 supplemented with 10% FBS and cells were cultured for an additional 24 h prior to treatment. The macrophages were then incubated in the complete RPMI 1640 for 18 h under the following experimental conditions a–h: (a) RPMI only (control); (b) IFNγ (100 units/mL); (c) HCQ (100 ng/mL); (d) HCQ (1000 ng/mL); (e) HCQ (10,000 ng/mL); (f) IFNγ + HCQ (100 ng/mL); (g) IFNγ + HCQ (1000 ng/mL); (h) IFNγ + HCQ (10,000 ng/mL). All incubations were done in triplicate. IFNγ dose is within the standard range for activation [20,21,22].

### 2.2. Cell Culture: Trypan Blue Viability Staining

Cells were centrifuged at 200 rcf for 5 min, the media was aspirated and the cells were resuspended in 0.4 mL of complete RPMI 1640 to give a cell density of at least 10^6^ cells/mL. Subsequently, 0.4 mL of syringe filtered 0.4% trypan blue (in PBS) was added directly to the cell suspension and thoroughly mixed. A hemocytometer was used to count the cells at 20X. Percent viability was then calculated as the ratio of viable (unstained) cells to the total number of cells.

### 2.3. RNA Isolation and QRT-PCR

Following 18–24 h of treatment, whole cell lysates were collected with intracellular RNA isolated using Trizol Reagent (Fischer Scientific, Waltham, MA, USA). RNA concentrations were quantified with the NanoDrop One (Fischer Scientific, Waltham, MA, USA), and standardized to a concentration of 1 μg/mL. RNA was then reverse transcribed to produce cDNA using the Mastercycler Nexus Gradient (Eppendorf, Hamburg, Germany). qPCR was performed using a Roche Lightcycler 480 system to quantify the levels of gene expression for key proteins responsible for macrophage cholesterol efflux: 27-hydroxylase (CYP27A1), ATP-binding cassette subfamily A-1 (ABCA1), and ATP-binding cassette subfamily G-1 (ABCG1), as well as scavenger receptors that facilitate cholesterol influx: cluster of differentiation 36 (CD36), and scavenger receptor A1 (SRA1) (Table 1). Each reaction was executed in triplicate and levels of gene expression were normalized to the housekeeping gene glyceraldehyde 3-phosphate dehydrogenase (GAPDH). Non-template controls were included for each primer pair to check for significant levels of any contaminants. A melting-curve analysis was performed to assess the specificity of the amplified PCR products. The data are presented as fold-change ±95% confidence interval (CI).

### 2.4. Foam Cell Formation Assay

THP-1 monocytes (250,000 cells/mL) were transferred into 8-well glass-chamber slides, and treated with 100 nm PMA (1 μL/mL) in complete RPMI 1640 for 18–24 h at 37 °C to stimulate differentiation into macrophages. THP-1 macrophages were then incubated for an additional 18–24 h at 37 °C under treatment conditions a–h as described above.

For the foam cell formation assay, differentiated macrophages were cholesterol-loaded with 25 µg/mL oxidized (ox)LDL (Intracel, Frederick, MD, USA) under the previously described treatment conditions a–h for an additional 18–24 h at 37 °C in the presence/absence of modified LDL.

For Oil Red O staining, the media was aspirated and the cells were washed with PBS and fixed in 4% paraformaldehyde for 10 min. They were washed with PBS for 1 min followed by a quick 15 s rinse with 60% isopropanol (Sigma). After rinsing, the cells were stained with 0.5% Oil Red O in 60% isopropanol for 1 min at 37 °C in the dark. The stain was removed by aspiration and cells were rinsed with 60% IPA for 15 s and 2 PBS washed for 3 min each. Cell nuclei were then stained using 0.4% Trypan Blue for 10 min. After a final wash with distilled water, coverslips were mounted on slides using Permount solution (Sigma).

Foam cells, recognized as macrophages stained with Oil Red O, were visualized via light microscopy (Axiovert 25; Carl Zeiss, Gottingen, Germany) with 40× magnification and photographed using a DC 290 Zoom digital camera (Eastman Kodak, Rochester, NY, USA). Number of foam cells formed in each condition was calculated in triplicate manually and presented as percentage of total cells.

### 2.5. OxLDL Uptake

OxLDL uptake was analyzed in THP-1 cells plated in 8 well chamber slides at 250,000 cells/mL in the presence of treatment conditions a-o described above for 18 h then incubated with 0.25 µg/mL 1,1′-dioctadecyl-3,3,3′,3′-tetramethylin docarbocyaninet (DiI)-oxLDL (Intracel, Frederick, MD, USA) for 4 h at 37 °C in the dark. Subsequently, cells were fixed in 4% paraformaldehyde for 10 min, washed with sterile PBS, and then prepared using Vectashield mounting medium containing DAPI stain (Vector Laboratories, Inc., Burlingame, CA, USA). After incubation, accumulation of DiI-oxLDL in cells was determined by fluorescent intensity with a Nikon A1 microscopy unit with 20× magnification and photographed with a DS-Ri1 digital camera. Fluorescent intensity was quantified from 9 random fields (1024 × 1024 pixels) per slide and maximum corrected total cell fluorescence (FU) was calculated.

### 2.6. Cholesterol Efflux Analysis

Cholesterol efflux was analyzed in THP-1 cells plated in 96 well plates at 1 × 10^6^ cells/mL. All conditions were plated in triplicates × 2 using the Amplex Red cholesterol assay (Molecular Probes, Eugene, OR, USA), according to the manufacturer’s protocol. Experimental conditions included THP-1 cells in the presence of a–h as described. Performing reactions in the presence and absence of cholesterol esterase, total (TC) and free (FC) cholesterol were analyzed. Cholesterol esters (CE) were estimated as the difference between TC and FC and the CE/FC ratio was calculated. Fluorescence was read at 585 nm and cholesterol efflux was expressed as percentage of fluorescence in efflux medium to total fluorescence of the cells and medium combined.

### 2.7. pH Assay

The Invitrogen pHrodo™ Red AM Intracellular pH Indicator kit (P35372) was utilized as indicated to conduct the pH assay. A solution of pHrodo™ AM Ester and PowerLoad™ concentrate were diluted into Live Cell Imaging Solution (LCIS) (Cat. no. A14291DJ) to make a staining solution. The media was removed from the cells and they were washed once with the LCIS. The LCIS was removed and the cells were incubated in the pHrodo™ AM Ester staining solution for 30 min at room temperature. After incubation, the cells were washed with LCIS and analyzed using appropriate excitation/emission maxima.

### 2.8. Protein Isolation and Western Blotting

Protein was collected from whole cell lysates that were harvested following 18–24 h of treatment using radioimmunoprecipitation assay (RIPA) lysis buffer (98% PBS, 1% Igepal, 0.5% sodium deoxycholate, 0.1% sodium dodecyl sulfate [SDS]), supplemented with 10 μL per ml of protease inhibitor cocktail (Sigma). Protein content was measured in triplicate using the BCA Protein Assay Kit by absorption at 562 nanometers (Pierce Biotechnology Inc., Rockford, IL, USA). 

Whole cell lysate protein extracts were separated and analyzed by 10% SDS-polyacrylamide gel electrophoresis (SDS-PAGE). 20 µg of each sample were loaded per gel lane and transferred to PVDF membranes. The blots were subsequently blocked with 5% NFDM in TBST for 1 h at room temperature and incubated in primary antibodies at 4 °C overnight. For immunoblot analysis, proteins were probed with Abcam primary antibodies against CYP27A1 (ab126785, rabbit monoclonal), ABCG1 (ab52617, rabbit monoclonal), CD36 (ab252922, rabbit monoclonal) and SRA1 (ab136802, rabbit polyclonal). All antibodies were diluted at 1:1000 concentrations in 5% NFDM in TBST. While a 1:1000 dilution of β-actin (Cell Signaling Technologies, CST3700, mouse monoclonal) in 5% NFDM in TBST was used as a control. Bound antibodies were visualized with 1:2000 dilution of their respective horseradish peroxidase-conjugated secondary antibodies prepared in 1% NFDM in TBST. The immunoreactive protein was detected using ECL western blotting detection reagent (Thermo Scientific™ SuperSignal™ West Pico PLUS Chemiluminescent Substrate) and the Bio-Rad ChemiDoc Touch Imaging System. Protein expression was normalized with respect to expression of β-actin and quantified via densitometry with ImageJ computer software (Rasband, 2018). The data are represented as mean ± standard error of the mean (SEM).

### 2.9. Statistical Analysis of Experimental Data

Statistical analyses were performed using GraphPad Prism, version 6, Microsoft Excel, and R Studio version 4.04. All normally distributed data were analyzed by factorial ANOVAs if fully balanced, otherwise data were analyzed using one-way ANOVA testing with pairwise multiple comparisons, or correlations. A *p*-value of < 0.05 was considered as statistically significant for all tests. Appropriate non-parametric equivalents (e.g., Kruskal–Wallis) were run on non-normally distributed data. Data points were excluded from PCR, Western blot, cholesterol uptake, and efflux analyses if there was no recovered RNA, low recovered protein, poor image quality, and inappropriate standard curves, respectively.

## 3. Results

### 3.1. Gene Expression–Efflux Genes

No significant differences were observed within the efflux genes ABCA1 and ABCG1 (Figure 1A,B, respectively; for HCQ *p* = 0.97 and 0.73, respectively. CYP27A1 expression in HCQ treatment groups also displayed no significant changes (Figure 1C; *p* = 0.85).

### 3.2. Gene Expression–Scavenger Receptors

Gene expression for the scavenger receptors, CD36 and SRA1, exhibited essentially no significant changes across all treatment groups (Figure 2A,B, respectively; for HCQ *p* = 0.77 and 0.79, respectively).

### 3.3. Protein Expression–Efflux Proteins

Both ABCA1 and ABCG1 protein expression displayed no significant effect. A representative blot for ABCG1 is shown in Figure 3.

Treatment with IFNγ significantly decreased CYP27A1 only in the HCQ-treated cells, indicating differential effects of IFNγ depending on the presence or absence of HCQ (F(6,28) = 2.969; *p* = 0.0226; interaction effect) (Figure 4).

### 3.4. Protein Expression–Scavenger Receptors

No significant differences were seen in CD36 expression across groups (F(6,42) = 0.719; *p* = 0.6367). SRA1 displayed similar results to CD36 with no changes in expression across groups (F(6,28) = 1.769; *p* = 0.1290).

### 3.5. Viability

Average cell viability was not affected by treatment with HCQ at any dose indicating that the drug was not toxic to these cells (*p* = 0.65).

### 3.6. Cholesterol Efflux

Using the Amplex Red assay, we quantified intracellular and supernatant cholesterol levels and showed that in THP-1 macrophages incubated in RPMI media alone, HCQ at increasing concentrations did not significantly change efflux of TC or FC and in cells incubated in RPMI media with 100 units/mL IFNγ, HCQ at increasing concentrations did not significantly change efflux of TC or FC (Figure 5).

### 3.7. Lipid Uptake, Staining and Foam Cells

For DiI and oxLDL uptake, no significant differences in cholesterol uptake were observed across all treatment groups (for HCQ *p* = 0.48) (Figure 6). Oil Red O staining was quantified and no significant differences were observed (*p* = 0.93) (Figure 7).

### 3.8. pH

No significant differences were observed in pH across all treatment groups (for HCQ *p* = 0.27).

## 4. Discussion

This study uses THP-1 human macrophages to evaluate effects of HCQ on cholesterol transport and accumulation as an indicator of possible atheroprotective or atherogenic properties of these drugs in RA patients. THP-1 macrophages share many qualities with primary human macrophages and have been used in numerous in vitro cell-based atherosclerosis studies by our group and others [23,24,25]. Excess lipoprotein uptake and impaired cholesterol efflux promote the transformation of macrophages into lipid-overloaded foam cells, which are characteristic of the early stage of atherosclerotic lesion development [26].

IFNγ is an atherogenic cytokine that is elevated in RA [27,28]. It downregulates cholesterol efflux gene expression and impairs macrophage cholesterol efflux, enhancing foam cell formation [29]. We therefore performed our studies with and without IFNγ to provide an inflammatory environment and stress the macrophages [30]. The use of IFNγ is common as a means of activating M1 macrophages that then produce TNF-α, and reactive oxygen species, both of which are associated with RA [31,32,33].

Our in vitro experiments show that HCQ did not significantly change lipid accumulation in THP-1 macrophages, either via augmented influx of cholesterol through scavenger receptors or attenuated outflow through efflux proteins. We chose to investigate the expression of these particular genes because there is a vast literature affirming their importance in cholesterol homeostasis and foam cell formation. Reverse cholesterol transport out of macrophages occurs in a coordinated fashion via ABCA1, ABCG1, and 27-hydroxylase [34,35,36,37,38]. Reverse cholesterol transport counters internalization of cholesterol via scavenger receptors, primarily SRA1 and CD36. These receptors take in modified forms of LDL, including oxidized and acetylated LDL, and are not feedback-inhibited in the presence of excess lipids [39,40]. Major characteristics of reverse cholesterol transport proteins and scavenger receptors are summarized in Table 2.

This neutral status of HCQ is relevant to TARGET and to clinical decision-making in the treatment of RA because ASCVD risk is such an important, often overlooked component of RA health consequences [41]. Although some studies show that HCQ improves lipid profile in RA patients, the impact of this change on actual heart health and cardiac event risk still remains in doubt [42,43]. Animal models have also shown attenuation of atherosclerosis by HCQ, but mouse models often do not correspond well to human findings [44]. Once TARGET data becomes available, we can evaluate the results of our study in context and, at that time, it may be possible to look at expression of cholesterol efflux proteins and scavenger receptors in peripheral monocytes collected from those patients.

## 5. Conclusions

Our in vitro examination of HCQ effects on THP-1 human macrophage lipid transport-related gene and protein expression and lipid handling does not provide a rationale for an atheroprotective role for this drug class in RA patients. Interestingly, a trial is underway exploring the effectiveness of HCQ in hospitalized patients with myocardial infarction. 2500 such patients will be randomized to HCQ or placebo for one year to determine whether HCQ will reduce recurrent cardiovascular events [45,46]. Our findings may be relevant to this trial as well. A study of HCQ in 19 premenopausal female patients with systemic lupus erythematosus found that the drug increased transfer of unesterified cholesterol to HDL, which would indicate improved HDL function with HCQ [47]. Clinical data combined with our type of cell culture study may be helpful in the future in understanding mechanisms and predicting success or failure of many drugs being considered for cardioprotection due to their anti-inflammatory properties.

## Figures and Tables

**Figure 1 medicina-58-01287-f001:**
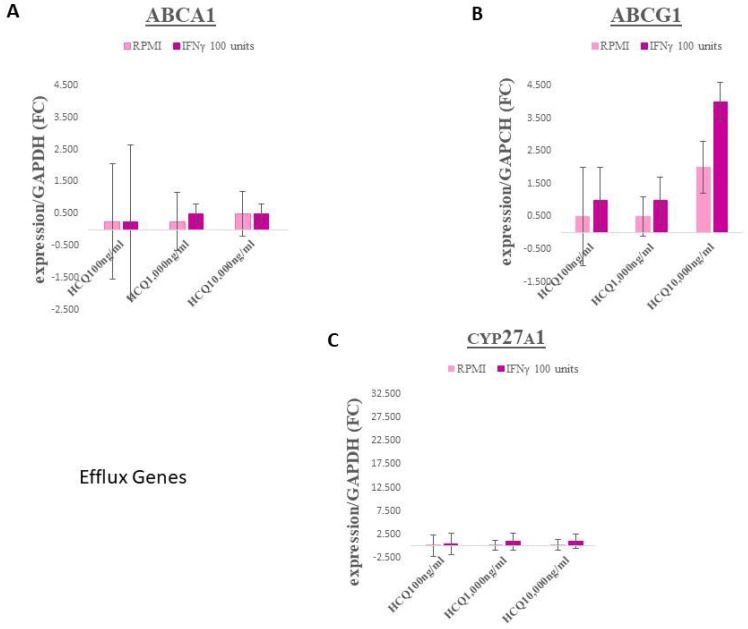
Effect of HCQ on the expression of cholesterol efflux genes in THP-1 macrophages. Following incubation, mRNA was isolated for each condition, reverse transcribed and amplified by QRT-PCR with GAPDH message as an internal standard. Values are fold-change ±95% CI relative to untreated control. (**A**) ABCA1, (**B**) ABCG1, (**C**) CYP27A1 (27-hydroxylase). Light pink bars = media, no IFNγ; Dark pink bars = IFNγ, 100 units/mL added to media. There were no significant interaction effects between IFNγ and HCQ in any genes.

**Figure 2 medicina-58-01287-f002:**
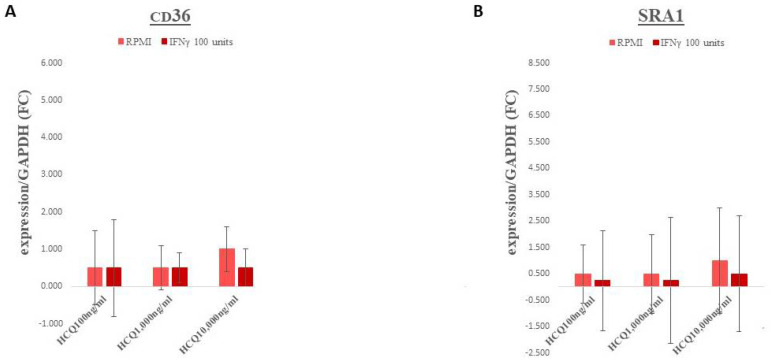
Effect of HCQ on the expression of Scavenger Receptor Genes in THP-1 macrophages. Following incubation, mRNA was isolated for each condition, reverse transcribed and amplified by QRT-PCR with GAPDH message as an internal standard. Values are fold-change ±95% CI relative to untreated control. (**A**) CD36 (**B**) SRA1. No significant interactions were observed in either CD36 or SRA1 expression. Light red bars = media, no IFNγ; Dark red bars = IFNγ, 100 units/mL added to media.

**Figure 3 medicina-58-01287-f003:**
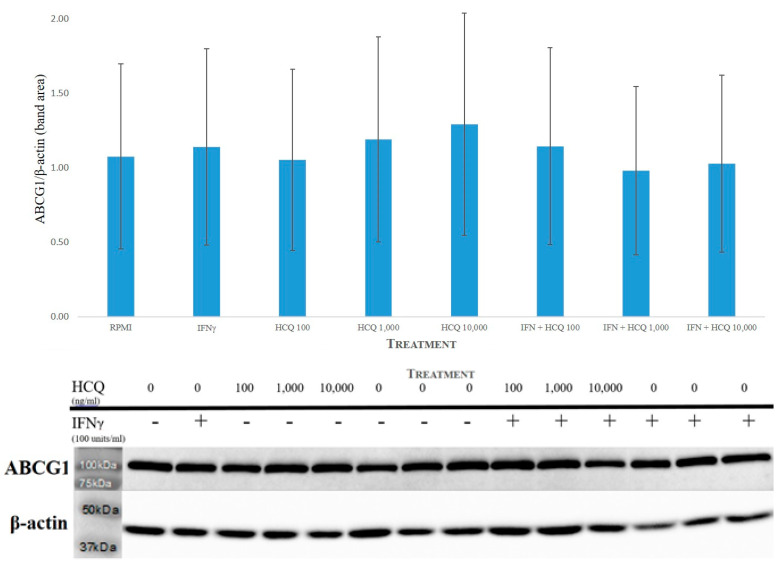
Effect of HCQ in the presence and absence of IFNγ on the expression of ABCG1. Western blot analysis using ABCG1 specific antibody after 18 h incubation of THP-1 macrophages. β-actin was detected on the same membrane as a loading control. **Top**: Results presented as a graph for ABCG1 protein expression with band densities normalized against β-actin. **Bottom**: Immunoblot with ABCG1 and ß-actin detected as immunoreactive bands. The data represent the mean and SEM of three independent experiments (*n* = 3). Western blot and densitometry analyses showed no significant interaction.

**Figure 4 medicina-58-01287-f004:**
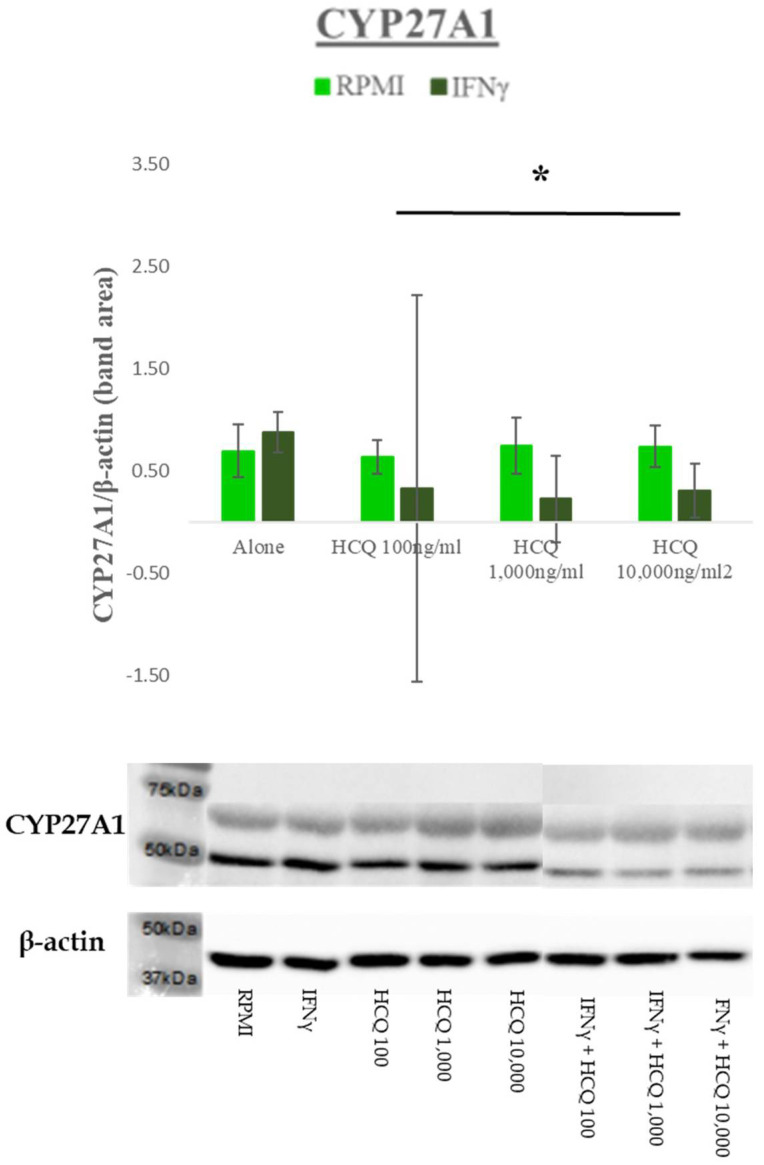
Effect of HCQ in the presence and absence of IFNγ on the expression of CYP27A1. Western blot analysis using CYP27A1 specific antibody after 18 h incubation of THP-1 macrophages. β-actin was detected on the same membrane as a loading control. **Top**: Immunoblot results presented as a graph for CYP27A1 protein expression with band densities normalized against ß-actin. **Bottom**: Immunoblot with CYP27A1 and ß-actin detected as immunoreactive bands. The data represent the mean and SEM of three independent experiments (*n* = 3). Western blot and densitometry analyses showed that CYP27A1 expression decreased with HCQ in the presence of IFNγ. Light green bars = media, no IFNγ; Dark green bars = IFNγ, 100 units/mL added to media. * *p* < 0.05 for the interaction effect only between IFNγ and HCQ.

**Figure 5 medicina-58-01287-f005:**
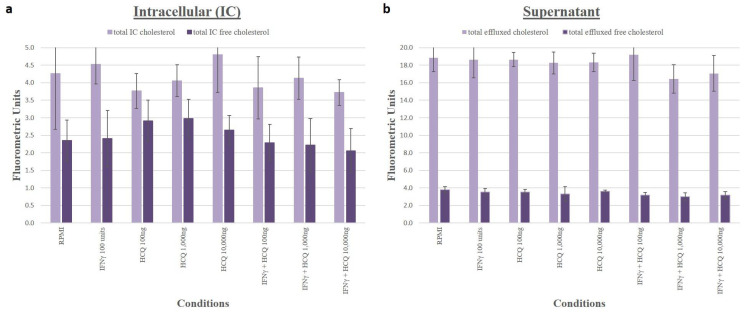
Cholesterol Efflux from Macrophages in Response to HCQ. Amplex Red assay showed no significant change in cholesterol efflux from cells to media in cells treated with HCQ compared to control (no HCQ) in the presence or absence of IFNγ. Cells exposed to HCQ had no change in total or free cholesterol efflux compared to RPMI and cells exposed to HCQ in the presence of IFNγ had no change in total or free cholesterol efflux compared to IFNγ alone. (**a**) Light purple bars = total intracellular cholesterol; Dark purple bars = intracellular free cholesterol. (**b**) Light purple bars = total supernatant cholesterol; Dark purple bars = supernatant free cholesterol. Values are mean fluorometric units +/− SD.

**Figure 6 medicina-58-01287-f006:**
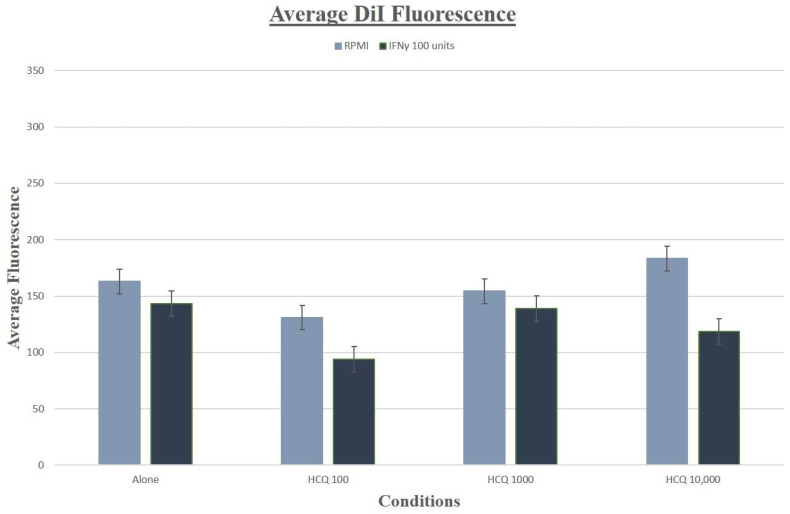
Effect of HCQ on oxLDL uptake into THP-1 Macrophages. DiI fluorometric staining and analysis did not demonstrate any significant effects on influx into cells treated with HCQ and IFNγ across conditions. Values are mean fluorometric units +/− SD. Light blue bars = RPMI media alone, no IFNγ; Dark blue bars = IFNγ, 100 units/mL added to media.

**Figure 7 medicina-58-01287-f007:**
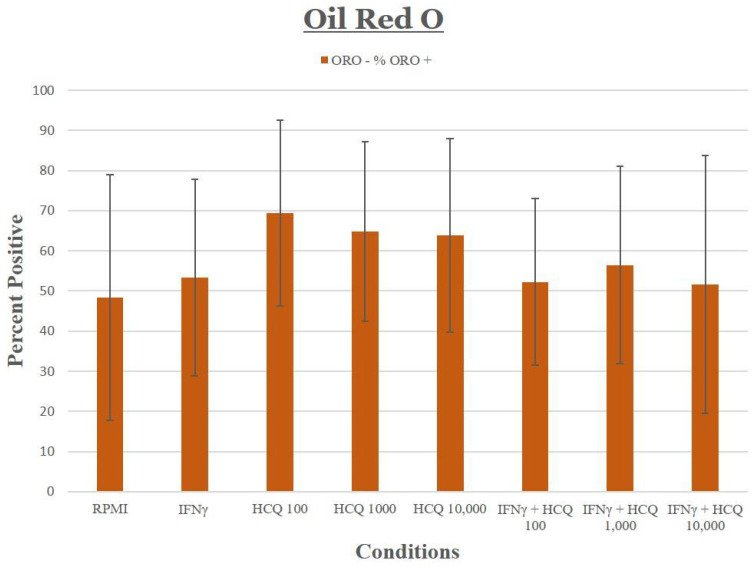
THP-1 macrophages were stained with Oil red O (ORO) to detect foam cells. Graphic representation of percentage foam cells shows no significant change in foam cell formation under cholesterol loading conditions with increasing concentrations of HCQ in the presence or absence of IFNγ.

**Table 1 medicina-58-01287-t001:** Forward and reverse primers used to measure expression of cholesterol efflux genes and scavenger receptors.

Gene	Sequence
ABCA1	Forward: 5′-GAAGTACATCAGAACATGGGC-3′ Reverse: 5′-GATCAAAGCCATGGCTGTAG-3′
ABCG1	Forward: 5′-CAGGAAGATTAGACACTGTGG-3′ Reverse: 5′-GAAAGGGGAATGGAGAGAAG-3′
CYP27A1	Forward: 5′-AAGCGATACCTGGATGGTTG-3′ Reverse: 5′-TGTTGGATGTCGTGTCCACT-3′
CD36	Forward: 5′-GAGAACTGTTATGGGGCTAT-3′ Reverse: 5′-TTCAACTGGAGAG-GCAAAGG-3′
SRA1	Forward: 5′-CTCGTGTTTGCAGTTCTCA-3′ Reverse: 5′-CCATGTTGCTCATGTGTTCC-3′
GAPDH	Forward: 5′-ACCATCATCCCTGCCTCTAC-3′ Reverse: 5′-CCTGTTGCTGTAGCCAAAT-3′

ATP-binding cassette subfamily A-1, ABCA1; ATP-binding cassette subfamily G-1, ABCG1; cholesterol 27-hydroxylase (cytochrome P450 family 27 subfamily A member 1), CYP27A1; cluster of differentiation 36, CD36; scavenger receptor A1, SRA1; glyceraldehyde 3-phosphate dehydrogenase, GAPDH.

**Table 2 medicina-58-01287-t002:** Characteristics of key cholesterol efflux proteins in THP-1 macrophages.

Name	Description
ABCA1	A full-size ABC transporter that mediates active efflux of cholesterol from macrophages and other nonhepatic cells to lipid-free apolipoprotein A-I (prevents foam cell formation). Essential for the generation of high density lipoprotein (HDL).
ABCG1	Mediates the removal of lipid molecules, including cholesterol and phospholipids, from macrophages and their transport across cellular and intracellular membranes to HDL particles.
Cholesterol 27-Hydroxylase (CYP27A1)	A mitochondrial enzyme that catalyzes the hydroxylation of cholesterol to more polar oxysterol metabolites that exit cells through lipid membranes orders of magnitude faster than cholesterol. It also regulates cholesterol biosynthesis via its major metabolite, 27-hydroxycholesterol, which potently inhibits cholesterol synthesis.
CD36	A member of the class B scavenger receptor family crucial for macrophage uptake of modified LDL, primarily oxidized LDL
SRA1	Expressed mainly by mature macrophages. Binds to oxidized LDL and other modified forms of LDL and mediates their internalization.

ATP binding cassette transporter, ABC; ATP binding cassette transporter A1, ABCA1; ATP binding cassette transporter G1, ABCG1; cluster of differentiation 36, CD36; cholesterol 27-hydroxylase, CYP27A1; high density lipoprotein, HDL; low density lipoprotein, LDL; scavenger receptor A1, SRA1.

## Data Availability

The data that support the findings of this study are available from the corresponding author upon reasonable request.

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
