# Peer review of "Hydroxychloroquine Effects on THP-1 Macrophage Cholesterol Handling: Cell Culture Studies Corresponding to the TARGET Cardiovascular Trial"

_medicina, 2022, doi:10.3390/medicina58091287_

Round 1

Reviewer 1 Report

The paper Medicina-1836079 studies THP-1 macrophage cholesterol handling by exposing macrophages to media alone, interferon gamma (IFNγ), hydroxychloroquine (HCQ,) or HCQ plus IFNγ.  Cholesterol efflux proteins and scavenger receptor mRNA levels were quantified by PCR and the corresponding protein levels were analyzed by Western blot. Foam cell formation was assessed by microscopy.   Intracellular cholesterol and cholesterol efflux were quantified by the Amplex Red assay.  HCQ did not significantly affect THP1 macrophage cholesterol transport. The only significant change the authors describe is a decrease in the cholesterol efflux protein CYP27A1 (27-hydroyxlase) in the presence IFNγ and HCQ. The authors conclude that HCQ did not significantly affect THP1 macrophage cholesterol transport.

This is a well-conducted in vitro study with essentially negative results, but of importance for understanding the potential of HCQ as an atheroprotective agent. I have two questions for the authors:

-        Please, explain Fig 1 B. With IFNγ and HCQ at 10,000 ng/ml the expression of ABCG1 is more than double than in medium (RPMI) plus IFNγ. The error bars do not overlap.  Was this change not significant?

-        Please, explain Fig 4. The expression of CYP7A1 in the presence of IFNγ and HCQ is decreased in comparison to the presence of medium (RPMI) +/- HCQ, but the error bars (hugely) overlap. Were any direct comparisons significantly different or was only the overall interaction effect different?

Minor point

Introduction, line 36: the refs. 6 and7 should be separated by a comma.

Author Response

Reviewer # 1 Comments

  • COMMENT#1: Please, explain Fig 1 B. With IFNγ and HCQ at 10,000 ng/ml the expression of ABCG1 is more than double than in medium (RPMI) plus IFNγ. The error bars do not overlap. Was this change not significant?

RESPONSE: We thank the reviewer for the positive assessment. Although there is a large difference between these conditions, it was part of a factorial analysis which was non-significant overall.  Therefore, those individual differences were not assessed separately from the overall analysis. We have added an explanatory sentence to the figure legend.

  • COMMENT#2: Please, explain Fig 4. The expression of CYP7A1 in the presence of IFNγ and HCQ is decreased in comparison to the presence of medium (RPMI) +/- HCQ, but the error bars (hugely) overlap. Were any direct comparisons significantly different or was only the overall interaction effect different?

RESPONSE: We apologize for being unclear - this is another interaction effect where there was only overall significance but no significant direct comparisons. We have added the following to the figure legend: “*p<0.05 for the interaction effect only between IFNγ and HCQ.”

  • COMMENT#3: Minor point: Introduction, line 36: the refs. 6 and7 should be separated by a comma.

RESPONSE: We have added the comma.

Reviewer 2 Report

The article exam HCQ effects on THP-1 macrophage choleterol handling by cell cultures. The study was well desgined and well performed and the conclusion was solid. 

Author Response

Reviewer # 2 Comments

  • COMMENT#1: The article exam HCQ effects on THP-1 macrophage cholesterol handling by cell cultures. The study was well designed and well performed and the conclusion was solid.

RESPONSE: We appreciate the affirmative evaluation of our manuscript.